# Changes in Saliva Analytes in Dairy Cows during Peripartum: A Pilot Study

**DOI:** 10.3390/ani11030749

**Published:** 2021-03-09

**Authors:** María D. Contreras-Aguilar, Pedro J. Vallejo-Mateo, Elsa Lamy, Damián Escribano, Jose J. Cerón, Fernando Tecles, Camila P. Rubio

**Affiliations:** 1Interdisciplinary Laboratory of Clinical Analysis (Interlab-UMU), Veterinary School, Regional Campus of International Excellence ‘Campus Mare Nostrum’, University of Murcia, 30100 Murcia, Spain; mariadolores.contreras@hotmail.com (M.D.C.-A.); det20165@um.es (D.E.); jjceron@um.es (J.J.C.); camila.peres@um.es (C.P.R.); 2Department of Animal Medicine and Surgery, Veterinary School, Campus Mare Nostrum, University of Murcia, 30100 Murcia, Spain; pedroja512@hotmail.com; 3MED—Mediterranean Institute for Agriculture, Environment and Development, IIFA—Instituto de Investigação e Formação Avançada, University of Évora, Núcleo da Mitra, Apartado 94, 7006-554 Évora, Portugal; ecsl@uevora.pt; 4Department of Animal Production, Veterinary School, Regional Campus of International Excellence ‘Campus Mare Nostrum’, University of Murcia, 30100 Murcia, Spain; 5Department of Veterinary Clinics, School of Veterinary Medicine and Animal Science, São Paulo State University (UNESP), Botucatu 18618-681, Brazil

**Keywords:** biomarkers, dairy cattle, peripartum, saliva, sialochemistry

## Abstract

**Simple Summary:**

The use of saliva as a biological fluid to assess welfare and health status is gaining interest nowadays since it can be collected by non-invasive methods without specialized staff. The possibility of measuring analytes in saliva by techniques adapted to automated analyzers is cost-effective, reliable, and replicable. These analytes can provide information useful for the evaluation of welfare and health in dairy cows. In this pilot study, a total of 26 salivary analytes were measured in healthy dairy cows along their peripartum period to assess possible changes and associations with their inflammatory, energy, and milk yield status. Salivary analytes related to stress (cortisol, salivary alpha-amylase, butyrylcholinesterase, and total esterase), immunity (adenosine deaminase), oxidative status (Trolox equivalent antioxidant capacity and the advanced oxidation protein products), and general metabolism (creatine kinase, γ-glutamyl transferase, urea, triglycerides, glucose, and lactate) had significant changes throughout this period. This study indicated that the saliva of dairy cows shows changes in its composition during the transition period and potentially can be a source of biomarkers for monitoring health and welfare.

**Abstract:**

This pilot study aimed to study the possible changes in a profile of 26 salivary analytes in thirteen healthy dairy cows along their peripartum period. Analytes associated with the stress (salivary cortisol, salivary alpha-amylase, butyrylcholinesterase, and total esterase), inflammation (adenosine deaminase), oxidative status (total antioxidant capacity and the advanced oxidation protein products), and general metabolism (creatine kinase, γ-glutamyl transferase, urea, triglycerides, glucose, and lactate) varied along the sampling times. A positive correlation between the white blood cells counts, and the lipase, Trolox equivalent antioxidant capacity, advanced oxidation protein products, and lactate levels in saliva were observed at the delivery. A linear association between selected salivary analytes at different sampling times and the milk yield after calving was observed. In conclusion, in our experimental conditions, it was observed that the peripartum period in dairy cows can induce changes in salivary analytes. Some of them were associated with inflammatory status and the capacity of milk production after calving.

## 1. Introduction

Nowadays, there is a growing number of studies in which saliva is being evaluated as a biomarker source since it contains biomolecules that change under some diseases or stress and inflammatory situations [1,2]. This is due to the fact that saliva can be easily collected by non-invasive and non-painful techniques, even at short-time intervals, and there is no need for specialized staff [3]. Recently, two studies where a biochemistry profile integrated by analytes that are usually measured in serum and plasma (called “sialochemistry”) was measured in the saliva of cows and reported changes in some of these analytes in lameness and mastitis [4,5].

The peripartum period in dairy cows, which is comprised approximately three weeks before and after calving, has a direct impact on their subsequent health status, production, and profitability. This period is considered the highest risk for developing infectious diseases and metabolic disorders and, therefore, compromised welfare, since they are submitted to a high metabolic challenge with significant physiological changes [6,7,8]. The welfare assessment in dairy cows can be made by behavioral (i.e., pain scales, stereotypes), physiological (i.e., heart rate variability), and performance (i.e., milk yield, fertility, dry matter intake, reduced body condition score-BCS-) parameters, but also by measurements in the blood hormone (i.e., cortisol, β-endorphin) and hematological and biochemical analytes (i.e., metabolites, leukocytes ratios, acute phase proteins, inflammatory products) [9]. However, to the best knowledge of these authors, there are no studies where the possible changes in salivary analytes have been evaluated during the peripartum period in dairy cows to assess whether the possible salivary analyte changes are in line with the previous changes reported in blood.

This study hypothesizes that the peripartum period would produce salivary changes in different analytes due to the physiological variations that dairy cows undergo during this period. Therefore, this study aimed to evaluate the changes in a profile of 26 analytes measured by automatized techniques in the saliva of dairy cows involving biomarkers of stress, immunity, oxidative status, and general metabolism during its transition period, from 3 weeks before to 3 weeks after calving.

## 2. Materials and Methods

Thirteen multiparous Holstein Friesian dairy cows (mean age = 4.7 ± 1.50; mean parity = 3.2 ± 1.46, min = 2, max = 7) in their last phase of gestation from a Spain commercial dairy herd (38°2′ N, 1°15′ W) were retrospectively selected according to a BCS > 3 at −20 ± 6.91 days relative to calving. In a previous sialochemistry study, ten cows gave adequate power [4]. An experienced veterinarian (PJV-M) checked all the cows, and no one showed any health issues (i.e., mastitis, metritis, ketosis, or lameness) and hematological and serum biochemical abnormalities in routine profiles. BCS was evaluated according to the visual five-point scale reported by Edmondson et al. [10] Samples were obtained between January and February of 2019 to avoid any change in the selected analytes due to seasonal reasons.

The feeding was based on a total mixed ration and was offered ad libitum at 08:30 am. The cows from −50 d relative to calving to pre-calving time received a far-off diet (1.49 Mcal/kg of dry matter, 9.4% rumen-degradable protein, and 5.3% rumen-undegradable protein) and were housed in small groups of 20–25 cows in sand-bedded free stalls (1.2 stalls/cow). Then, after 30 days post-calving, the cows were fed with a lactation diet (1.71 Mcal/kg of dry matter, 11.0% rumen-degradable protein, and 6.0% rumen-undegradable protein) and were housed in other free-stalls (1.1 stalls/cow) with straw. At this time, the cows were checked daily and milked two times a day. Water intake was available ad libitum. The farm was free of brucellosis, tuberculosis, Bovine Leukosis Virus, and pleuropneumonia. The cows were vaccinated for Bovine Viral Diarrhea, Infectious Bovine Rhinotracheitis, Bovine parainfluenza-3, and Bovine respiratory syncytial virus.

The saliva sampling was performed by introducing a sponge clipped to a flexible thin metal rod into the cow’s mouth. The sponge was then placed in a collection tube Salivette (Sarstedt, Aktiengesellschaft & Co., Nümbrecht, Germany). After that, blood was obtained by venipuncture in the vena caudal, and it was stored in a heparin device (LH/Li Heparin, Aquisel, Barcelona, Spain). The saliva and blood were obtained in the milking parlor when the milkers were removed and the nipples were post-dipped [4]. Cows were previously made accustomed to the procedure for saliva and blood sampling. Both saliva and blood devices were stored less than two hours in ice until the processing: the saliva was centrifuged at 3000× *g* for 10 min at 4 °C, and the blood analyzed for the white blood cell (WBC) count. Saliva specimens were stored for less than five months at −80 °C until analysis.

The salivary analytes evaluated in the chemistry profile were:Stress biomarkers. Salivary cortisol (sCor), salivary alpha-amylase (sAA), butyrylcholinesterase (BChE), total esterase (TEA), and lipase (Lip) [11,12].Immunity biomarkers. Adenosine deaminase (ADA) [13].Oxidative status biomarkers. Trolox equivalent antioxidant capacity (TEAC), the ferric reducing ability of saliva (FRAS), the cupric reducing antioxidant capacity (CUPRAC), uric acid, advanced oxidation protein products (AOPP), and hydrogen peroxide (H_2_O_2_) [14,15].Biomarkers of general metabolism and liver, muscle, and renal damage (enzymes, proteins, and minerals). Aspartate aminotransferase (AST), alanine aminotransferase (ALP), γ-glutamyl transferase (gGT), lactate dehydrogenase (LDH), creatine kinase (CK), creatinine, urea, triglycerides, glucose, lactate, total protein, albumin, phosphorus, and total calcium.

This salivary chemistry profile was analyzed in an automated chemistry analyzer using adapted kits to this fluid previously reported and validated [4,5,14,16]. The cortisol in saliva was analyzed according to Escribano et al., [17]. The WBC count was evaluated using an automated analyzer (ADVIA 120 Hematology System, Siemens Healthcare, Spain).

Saliva and blood samplings were carried out at −13 ± 6.91 days relative to calving (T − 13), within the 12 first hours after calving (T0), and seven days (T + 7) and 20 days (T + 20) after calving. Samples were obtained in the same period each day (from 10:30 am to 12:30 pm). The diagnostic of gestation was performed by rectal examination and ultrasounds at least 32 days post-insemination. Changes in the BCS was assessed as previously indicated. Milk yield and cow activity were evaluated as Contreras-Aguilar et al. described [4].

Firstly, the analytes showing a non-normal distribution using a Kolmogorov-Smirnov test were log-transformed [18] to restore normality. Data for all measured variables (salivary analytes, WBC count, milk yield, and BCS) were analyzed as repeated measures using the mixed linear model of SPSS in which time was considered as fixed factor, the BCS and the nested interaction on parity × age as random factors, and the BCS at T − 13 as covariate. Each analyzed variable was subjected to three covariance structures: compound symmetric, autoregressive order one, and unstructured covariance. The covariance structure with the lowest Akaike’s information criterion and Schwarz’s Bayesian criterion was considered the most desirable analysis [19]. The relationship between each salivary analyte at each time with the WBC count, milk yield, and BCS in the time of changing were assessed for correlation using Pearson correlation coefficients. Those salivary analytes being significantly correlated with the WBC count, milk yield, and BCS were then assessed by multiple linear regression models (milk yield) or multiple logistic regression analyses (WBC count [4.1–11.9 × 10^9^ cel/L = 1 and ≤4.0 or ≥12.0 × 10^9^ cel/L = 2 [20]], and BCS (no change = 1 and decreasing at T20 compared to T − 13 = 2). The age and/or parity were included as predictor variables in the multiple logistic regression analyses only if they showed an effect in the mixed linear model for each analyte. The statistical analyses were performed using the SPSS statistics package (IBM SPSS statistics version 24 for Mac, IBM Corporation, Arkmon, NY, USA). Figures were represented using the GraphPad Prism 9 statistics package (GraphPad Software, La Jolla, CA, USA). Significance was set for *p* < 0.05.

## 3. Results

### 3.1. Salivary Chemistry Profile Changes

The plots from salivary analytes with significant changes are shown in Figure 1. Data of the salivary chemistry profile changes are shown in the Appendix A.

The salivary stress biomarkers that showed significant changes with time were sCor, sAA, BChE, and TEA. Salivary cortisol and BChE showed higher values at calving (T0) compared to before calving (T − 13; 1.49-fold, *p* = 0.023; 1.27-fold, *p* = 0.030), and seven (T + 7; 1.97-fold, *p* < 0.001; 1.53-fold, *p* < 0.001) and twenty (T + 20; 2.45-fold, *p* = 0.002; 1.35-fold, *p* = 0.012) days after calving. Salivary alpha-amylase activities were higher at T + 20 (1.71-fold, *p* = 0.040) compared to T − 13, while the TEA activities were higher at T − 13 (1.40-fold, *p* = 0.028) and T + 0 (1.25-fold, *p* = 0.028) compared to T + 7. Adenosine deaminase increased at T0 from T − 13 (4.90-fold, *p* = 0.010) decreasing then at T + 20 (−4.04-fold, *p* = 0.011).

The antioxidants TEAC (−6.00-fold, *p* = 0.002), FRAS (−4.05-fold, *p* = 0.008), and CUPRAC (−6.71-fold, *p* = 0.002) showed lower values at T0 compared to T − 13, as well as the AOPP (−3.17-fold, *p* = 0.050). The values of the enzymes CK (2.38-fold, *p* = 0.042) and gGT (2.59-fold, *p* = 0.004) were higher at T0 compared to T − 13, and in the case of gGT compared to T + 7 (−2.34-fold, *p* = 0.023) and T + 20 (−2.69-fold, *p* < 0.001). Concerning the metabolites, urea (−1.70-fold, *p* = 0.005), triglycerides (−3.86-fold, *p* = 0.036) and lactate (−7.81-fold, *p* = 0.006) showed lower values at T + 7 compared to T − 13, and glucose (−10.00-fold, *p* = 0.016) and lactate (−13.16-fold, *p* < 0.001) at T0 in relation to T − 13. Urea values were also lower at T + 7 (−1.38-fold, *p* = 0.026) compared to T0.

### 3.2. WBC Count, Milk Yield and BCS Changes

Cows showed a significant increase in the WBC count at T0 (*p* = 0.004) compared to T − 13, and in the milk yield at T7 (*p* < 0.001) and T20 (*p* < 0.001) with respect to T0 (Table 1). Otherwise, a decrease in the BCS was observed at T20 (*p* = 0.018) with respect T0.

### 3.3. Relationship between Salivary Analytes and WBC Count, Milk Yield and BCS

The correlation results are shown in Appendix A. The multiple logistic regression (Table 2) showed that Lip, lactate, TEAC, and AOPP showed association with the WBC values out of normal range at T0, whereas no significant associations were observed with a decreased BCS at T + 20 (Table 2).

The multiple linear regression models (Table 3) showed that the milk yield at T + 7 had a positive linear relationship with the values of sAA, CUPRAC, AOPP, H_2_O_2_, AST, lactate, and creatinine at T + 20, and the values of albumin at T + 7; and a negative linear relationship with the values of TEA and uric acid at T0. A negative linear relationship was observed between the milk yield at T + 20 and uric acid, AST, CK, creatinine, and urea at T0 (Table 3).

## 4. Discussion

In our experimental conditions, the decreases in BCS after the delivery in cows with high BCS before calving, and the rapid increases around the calving of blood polymorphonuclear leukocytes, decreasing transiently the first week after calving, are in line with the expected changes in this period as having been previously reported [19,21,22]. Besides, the WBC count observed on the day of calving compared to prepartum are similar to those obtained by Zachut et al. [23] in their cows during winter. Therefore, the healthy dairy cow population selected for this study showed the changes expected in this period regarding their BCS and leukocyte response.

In our study, various salivary stress biomarkers such as sCor, sAA, BChE, and TEA showed higher values the day of calving compared to prepartum and/or postpartum. Increases of the glucocorticoids and catecholamines around the time of calving by the hypothalamic–pituitary–adrenal axis and sympathetic nervous system (SNS) activation are associated with the delivery mechanisms [22,24] and could produce increases in cortisol and sAA [25]. BChE is also associated with the autonomic nervous system (ANS) in human beings [26] and increases in acute stress situations in pigs [27,28], sheep [11], and horses [12]. Salivary TEA, which also associated to the SNS activation [29], showed increases around thirteen days pre-calving and the day of calving, with median values similar to those reported in cows suffering lameness [4].

The ADA activity in saliva increased at calving. ADA activity has been associated with the immune system activation [30], with higher activities in T lymphocytes and macrophages [31], but also with the modulation of the inflammatory response regulating the extracellular levels of adenosine [32]. Therefore, increases in ADA activity could reflect the proinflammatory status in cows at calving [13,32].

The decreases in the salivary total antioxidant capacity measured by the TEAC, FRAS, and CUPRAC found in our study at calving and postpartum have been described in a previous study in the serum of cows [19,22], and might be due to different metabolic and endocrine mechanisms related to the fetus and mammary gland [19].

Creatinine kinase, gGT, urea, triglycerides, glucose and lactate showed significant changes during the transition period. The increases in CK and gGT could be related to the possible increases in blood due to muscle damage during the delivery [33] and the reduction of liver function usually observed in the early postpartum [6,8,18], respectively. However, they could be present in saliva due to a local salivary gland production, probably due to pain, as described in horses [16]. Reductions in the triglycerides and glucose concentrations in blood have been described during the post-partum period [34,35]. Similarly, urea remains low seven days post-partum [6] since the liver has a decreased capacity for urea synthesis due to the hepatocyte’s triglycerides accumulation, as can be observed in our study in saliva.

A correlation between altered values of WBC in blood and Lip, TEAC, AOPP, and lactate in saliva was detected on the day of calving. Recent research reported increases in salivary Lip in lameness dairy cows, indicating a possible relationship between this analyte in saliva and inflammatory or pain conditions [4]. AOPP might be a by-product of neutrophil activation being associated with inflammation [36]. Concerning the lactate in saliva, it is reported that lactate in the blood can come from muscle tissue or the mammary gland, modifying the inflammation affecting T cells and neutrophils function [37]. Besides, lactate concentration increases in the blood have been observed in dairy cows with retained placenta [37] and lameness [38] and in saliva from cows with mastitis [5]. From the 26 salivary analytes assessed at different sampling times, twelve analytes related to different situations such as pain-related stress, oxidative status, and metabolism had a linear association with the milk yield seven and twenty days after calving. Future studies should be undertaken to evaluate the possible use of these analytes as possible biomarkers of milk yield.

The results obtained in this study should be interpreted carefully due to some limitations. Only dairy cows from a particular breed and farm were enrolled, and no differences between cows of a different energy status or with different handling patterns (i.e., housed, milking, or nutrition handling) were taken into account. Therefore, other breeds and farming conditions should be tested. Otherwise, ideally the BCS would be assessed at least three times before calving to classify the cows correctly. In addition, a higher population should be evaluated to confirm these results and to assess possible differences in the sialochemistry profile between the healthy cows and dairy cows with common disorders observed during this period (e.g., ketosis, milk fever, retained fetal membranes and metritis, and displaced abomasum) [7]. Furthermore, it would be of interest to study the possible correlations between saliva and blood analytes; although, in a previous study in cows with mastitis, no evident correlations were found [5]. These additional studies in a large population and different farm conditions and diseases could elucidate whether the salivary analytes studied in this paper could be of interest as biomarkers for use in different situations, such as in the prevention or detection of selected diseases (i.e., in case of fatty acid mobilization imbalance) or situations of poor welfare or stress.

## 5. Conclusions

In this pilot study, the peripartum period in dairy cows induced changes in a profile of salivary analytes that could be associated with stress, changes in oxidative status or metabolism, and also with inflammation. The most striking changes were found in sAA, BChE, TEA, Lip, ADA, TEAC, FRAS, AOPP, gGT, and lactate. In addition, some salivary analytes at calving could be related to the milk yield. Therefore, this study indicated that saliva shows changes in its composition during the transition period in dairy cows and potentially could be a source of biomarkers for monitoring cows’ welfare and health.

## Figures and Tables

**Figure 1 animals-11-00749-f001:**
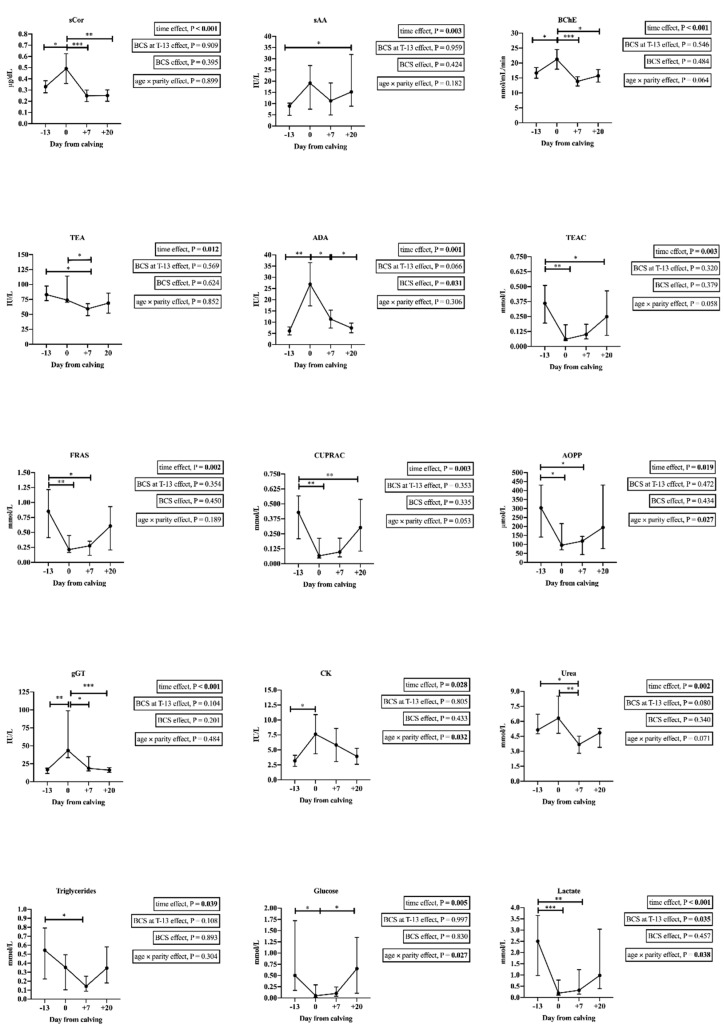
Results of salivary cortisol (sCor), salivary alpha-amylase (sAA), butyrylcholinesterase (BChE), adenosine deaminase (ADA), Trolox equivalent antioxidant capacity (TEAC), ferric reducing ability of saliva (FRAS), cupric reducing antioxidant capacity (CUPRAC), advanced oxidation protein products (AOPP), γ-glutamyl transferase (gGT), creatine kinase (CK), urea, triglycerides, glucose, and lactate in saliva of thirteen healthy dairy cows during their peripartum period (T − 13: at −13 ± 6.91 days relative to calving; T0: within the 12 first hours after calving; T + 7: seven days after calving; T + 20: 20 days after calving). The plots show the mean and 95% confidence interval (CI) (sCor, BChE, ADA, and CK) or the median and interquartile ranges (25–75%) (sAA, TEA, TEAC, FRAS, CUPRAC, AOPP, gGT, urea, triglycerides, glucose, and lactate). Asterisk indicates the statistically significant differences between times (Bonferroni pairwise comparison) (*: *p* < 0.05; **: *p* < 0.01; ***: *p* < 0.001). Time, body condition score (BCS) at T − 13 as factors and BCS and age × parity covariable effects were evaluated by a mixed linear regression model.

**Table 1 animals-11-00749-t001:** White blood cell (WBC) count, milk yield and body condition score (BCS) changes during the transition period. Values were expressed as mean (standard deviation).

	WBC Count (×10^9^ cel/L)	Milk Yield (kg/day)	BCS (Score)
Time (days from calving)			
−13	6.69 (1.44)	ND ^1^	3.4 (0.65)
0	10.74 (4.12) ^a^	3.7 (2.43)	3.5 (0.63)
+7	7.15 (2.17)	35.5 (6.87) ^b^	3.0 (0.29)
+20	7.81 (2.97)	38.5 (10.94) ^b^	2.9 (0.14) ^c^
Mixed linear regression model			
sampling time (F)	8.55 **	101.05 ***	4.79 **
BCS at T − 13 (F)	2.75	0.08	
BCS (Wald Z)	0.52	0.61	
Parity × age (Wald Z)	0.32	0.84	2.34 *

^1^ ND = no data. ^a,b,c^ Show statistical analysis results in Bonferroni pairwise comparison; ^a^
*p* = 0.004 with T − 13; ^b^
*p* < 0.001 with T − 13; ^c^
*p* = 0.018 with T − 13. * Asterisk indicates statistically significant difference (Mixed lineal regression model); (* *p* < 0.05; ** *p* < 0.01; *** *p* < 0.001).

**Table 2 animals-11-00749-t002:** Multiple logistic regression analyses in 13 dairy cows during their peripartum period between the altered white blood cell (WBC) at T0 (≤4.0 or ≥12.0 × 10^9^ cel/L [20]), and decreases in the body condition score (BCS) at T + 20 compared to T − 13 with some salivary analytes at different time of samplings. The times were T − 13: at −13 ± 6.91 days relative to calving; T0: within the 12 first hours after calving; T + 7: seven days after calving; T + 20: 20 days after calving. The analytical variable in bold showed significance.

Salivary Analytical Variable	Altered Value of WBC at T0		Salivary Analytical Variable	Decreases of the BCS at T + 20 Compared to T − 13	
	Crude Odds Ratio ^1^ (95% CI^2^)	*p* Value		Crude Odds Ratio (95% CI)	*p* Value
TEA ^3^ T0	1.96 (0.53–7.32)	0.315	sAA^10^ T + 20	0.66 (0.17–2.64)	0.561
**Lip ^4^ T0**	7.99 (1.01–64.82)	**0.050**	AST T0	2.97 (0.52–17.08)	0.223
**TEAC ^5^ T0**	17.22 (1.00–297.12)	**0.050**	CK T0	1.74 (0.28–10.80)	0.550
**AOPP ^6^ T0**	12.66 (1.02–156.93)	**0.048**	creatinine T0	11.63 (0.50–271.26)	0.127
H_2_O_2_ ^7^ T0	7.01 (0.72–68.36)	0.112	urea T0	1.07 (0.30–3.80)	0.921
AST ^8^ T0	5.08 (0.64–40.54)	0.125	uric acid T0	2.27 (0.61–8.51)	0.225
CK ^9^ T0	9.43 (0.78–114.99)	0.077	phosphorus T0	0.52 (0.12–2.22)	0.378
**lactate T0**	4.63 (1.13–18.98)	**0.033**	phosphorus T + 20	0.57 (0.14–2.36)	0.437
total calcium T0	6.09 (0.88–42.05)	0.067			

^1^ Crude odds ratio from the multiple logistic regression analyses. ^2^ CI = confidence interval. ^3^ TEA = total esterase. ^4^ Lip = lipase. ^5^ TEAC = trolox equivalent antioxidant capacity. ^6^ AOPP = advanced oxidation protein products. ^7^ H_2_O_2_ = hydrogen peroxide. ^8^ AST = aspartate aminotransferase. ^9^ CK = creatine kinase. ^10^ sAA = salivary alpha-amylase.

**Table 3 animals-11-00749-t003:** Multiple linear regression analyses in 13 dairy cows during their peripartum period between the dependent variable milk yield at T + 7 and T + 20, and some salivary analytes at different time of samplings as independent variables. The times were T − 13: at −13 ± 6.91 days relative to calving; T0: within the 12 first hours after calving; T + 7: seven days after calving; T + 20: 20 days after calving. The analytical variable in bold showed significance.

Salivary Analytical Variable	Milk Yield at T + 7			Milk Yield at T + 20			
	Linear Regression Equation	*R* ^2 1^	*p* Value	Salivary Analytical Variable	Linear Regression Equation	*R* ^2^	*p* Value
**sAA ^2^ at T + 20**	*y* = 4.99*x* + 21.58	0.256	**0.045**	**AST at T0**	*y* =−0.53*x* + 51.03	0.416	**0.014**
**TEA ^3^ at T0**	*y* =−13.11*x* + 93.69	0.418	**0.01**	gGT ^9^ at T + 20	*y* =−2.36*x* + 45.24	0.007	0.79
**TEA at T + 20**	*y* =15.28*x* − 28.08	0.258	**0.044**	**CK ^10^ at T0**	*y* =−1.38*x* + 49.55	0.422	**0.013**
**CUPRAC ^4^ at T + 20**	*y* =22.72*x* + 29.46	0.248	**0.048**	**creatinine at T0**	*y* =−41.95*x* + 69.19	0.589	**0.004**
**AOPP ^5^ at T + 20**	*y* =4.70*x* + 10.99	0.351	**0.019**	**uric acid at T0**	*y* =−109.17*x* + 64.34	0.51	**0.005**
**H_2_O_2_^6^ at T + 20**	*y* =5.01*x* + 16.24	0.38	**0.015**	**urea at T0**	*y* =−18.98*x* + 95.96	0.334	**0.029**
**AST ^7^ at T + 20**	*y* =0.80*x* + 25.91	0.31	**0.027**				
LDH ^8^ at T0	*y* =−0.06*x* + 40.28	0.112	0.141				
**creatinine at T + 20**	*y* =20.68*x* + 23.48	0.322	**0.025**				
**uric acid at T0**	*y* = −51.70*x* + 47.31	0.279	**0.037**				
**lactate at T + 20**	*y* = 9.44*x* + 28.00	0.452	**0.007**				
**albumin at T + 7**	*y* = −544.05*x* + 6.96	0.377	**0.015**				

^1^ Coefficient of determination; ^2^ sAA = salivary alpha-amylase; ^3^ TEA = total esterase; ^4^ CUPRAC = cupric reducing antioxidant capacity; ^5^ AOPP = advanced oxidation protein products; ^6^ H_2_O_2_ = hydrogen peroxide; ^7^ AST = aspartate aminotransferase; ^8^ LDH = lactate dehydrogenase; ^9^ gGT = γ-glutamyl transferase; ^10^ CK = creatine kinase.

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
