# Peer review of "Changes in Saliva Analytes in Dairy Cows during Peripartum: A Pilot Study"

_animals, 2021, doi:10.3390/ani11030749_

Round 1
Reviewer 1 Report
Very interesting manuscript regarding salivary concentration of several metabolites/biomarkers during the transition period of dairy cows. Nonetheless several major concerns arise.
- There is a lack of relevance in relation to which is the real aim of this manuscript. Is it to analyze several metabolites/markers in the saliva of transition cows or does it have a real aim at confirming previously reported changes during the transition period in the saliva? Several of the measured things have been measured in the plasma of transition cows. No mention to why so many things are being measured and the impact they would have in the management of the transition cow (Ej. disease prevention, fatty acid mobilization, behavioral changes?).
- I believe my previous point is relevant, considering that the transition period is a very complex period in which several modification in the "normal" physiology of the dairy cow get modify. This include fatty acid mobilization and behavioral changes that would directly impact your results and discussion. BCS scoring is very subjective and usually must be evaluated several times (at least 3 times before calving). Why did you use WBC to perform your correlations? This is even a more subjective and less accurate marker of anything, especially with only 1 sample.
- The M&M section is basically a carbon copy of your previous papers. I recommend al least modify some things.
- Nutrition is such an important factor that must be taking into account. Where did cows spend their pre calving time?..what where the conditions?. The factor of using only 1 precalving measurement is not good.
- My main concern, notheless is the statistical approach used. Although I'm perfectly fine with the used methods, I would like to point several aspects. A mixed model must have a random aspect (not mentioned here). Why did you use BCS, BCS*time, age and parity as covariates?...I believe they were factors in your model too?. I suggest to include the precalving value (only 1) as covariate. Similarly, probably, the parity nested in age and nested in the cow (parity/age/cow) must be considered the random factor. This will exclude most of your bias in your results.
- You should not compare your results to your WBC, which does not represents anything. I would suggest removing that section.
- I believe this would change your results and therefore your discussion.
- Have other smaller things, but would like to see this aspects modify first.
Author Response
Reviewer 1
Very interesting manuscript regarding salivary concentration of several metabolites/biomarkers during the transition period of dairy cows. Nonetheless several major concerns arise.
There is a lack of relevance in relation to which is the real aim of this manuscript. Is it to analyze several metabolites/markers in the saliva of transition cows or does it have a real aim at confirming previously reported changes during the transition period in the saliva? Several of the measured things have been measured in the plasma of transition cows. No mention to why so many things are being measured and the impact they would have in the management of the transition cow (Ej. disease prevention, fatty acid mobilization, behavioral changes?).
Answer: Thanks for this comment. This study has two aims: 1) analyze several metabolites/markers in the saliva of transition cows, but also 2) evaluate whether those salivary analytes changes are in line with previously reported changes during the transition period in blood. Therefore, to reflect this, the authors have added some changes in the “Introduction” section and now it can be read like this (lines 71-82):
“ (…) However, to the best knowledge of these authors, there are no studies where the possible changes in salivary analytes have been evaluated during the peripartum period in dairy cows to assess whether the possible salivary analytes changes are in line with the previous changes reported in blood.
This study hypothesizes that the peripartum period would produce salivary changes in different analytes due to the physiological variations that dairy cows undergo during this period. Therefore, this study aimed to evaluate the changes in a profile of 26 analytes measured by automatized techniques in the saliva of dairy cows involving biomarkers of stress, immunity, oxidative status, and general metabolism during its transition period, from 3 weeks before to 3 weeks after calving”
Additionally, the following sentence has been added in the “Discussion” section (lines 348-352): “These additional studies in a large population and different farm conditions and diseases could elucidate if the salivary analytes studied in this paper could be of interest as biomarkers for use in different situations such as the prevention or detection of selected diseases (i.e., in case of fatty acid mobilization imbalance) or situations of poor welfare or stress.”
I believe my previous point is relevant, considering that the transition period is a very complex period in which several modification in the "normal" physiology of the dairy cow get modify. This include fatty acid mobilization and behavioral changes that would directly impact your results and discussion. BCS scoring is very subjective and usually must be evaluated several times (at least 3 times before calving).
Answer: We agree with the reviewer that the BCS could be considered a subjective method of assessing the amount of metabolizable energy stored in fat and muscle [1]. However, some papers in specific dairy cows journals describe only one BCS scoring to select high BCS cows before calving [2,3] retrospectively. In fact, those studies have reported differentiated fatty acid mobilization during the peripartum period between groups classified with different BCS (low, medium, and high), and, therefore, different metabolic changes after calving. In our study, the BCS did not change at -13 days before calving compared to at -20 ± 6.91 days relative to calving. The BCS evaluation was also made with a standardized and validated score [1] by a trained veterinarian scored.
Nevertheless, we have included this reviewer’s concern in the limitation paragraph from the “Discussion” section (lines 340-342): “Otherwise, ideally the BCS would be assessed at least three times before calving to classify the cows correctly.”
Why did you use WBC to perform your correlations? This is even a more subjective and less accurate marker of anything, especially with only 1 sample. You should not compare your results to your WBC, which does not represents anything. I would suggest removing that section. I believe this would change your results and therefore your discussion.
Answer: We used the WBC count because leukocytes measurements have been also assessed in other studies [4,5] for being considered an inflammatory marker and the first defense line against invading pathogens such as in mammary glands and endometrium in dairy cows. In addition, their number in the peripartum period are also conditioned by the stress and the energy mobilizing status [6–8]. Thus, if possible, we would like to keep showing the WBC results.
The M&M section is basically a carbon copy of your previous papers. I recommend al least modify some things.
Answer: We have modified this section. These changes modified the single similarity rate of our previous paper from 9% to less than 5% as the editorial office required us
Nutrition is such an important factor that must be taking into account. Where did cows spend their pre calving time?..what where the conditions?. The factor of using only 1 precalving measurement is not good.
Answer: We agree with the reviewer concerning the critical factor that nutrition has in the dairy cows’ health status and metabolic changes. The cows from their -50 d relative to calving only receive a far-off diet and are housed in the same stall in groups of 20-25 cows. When they calve, each calf is separated from the cow and is housed on another stall with other cows with their similar situation. Thus, the nutritional factor and housing conditions at the pre-calving time (T-13) were homogenous in order to not be a source of variation.
In order to clarify and include the reviewer concerns in the manuscript, we have made the following changes:
- To make the paragraph concerning the feed and housed conditions clearer (lines 102-113): “The feeding was based on a total mixed ration and was offered ad libitum at 08:30 am. The cows from -50 d relative to calving to pre-calving time received a far-off diet (1.49 Mcal/kg of dry matter, 9.4% rumen-degradable protein, and 5.3% rumen-undegradable protein) and were housed in small groups of 20-25 cows in sand-bedded free stalls (1.2 stalls/cow). Then, once calving to 30 days post-calving, the cows were fed with a lactation diet (1.71 Mcal/kg of dry matter, 11.0% rumen-degradable protein, and 6.0% rumen-undegradable protein) and were housed in other free-stalls (1.1 stalls/cow) with straw. At this moment, the cows were checked daily and milked two times a day. Water intake was available ad libitum.”
- In the limitation’s paragraph, we had highlighted the importance of the nutritional factor and the need to clarify its possible effects on salivary analytes (lines 336-340): “The results obtained in this study should be interpreted carefully due to some limitations. Only dairy cows from a particular breed and farm were enrolled, and no differences between cows at different energy status or with different handling patterns (i.e., housed, milking, or nutrition handling) were taken into account. Therefore, other breeds and farming conditions should be tested.”
My main concern, notheless is the statistical approach used. Although I'm perfectly fine with the used methods, I would like to point several aspects. A mixed model must have a random aspect (not mentioned here). Why did you use BCS, BCS*time, age and parity as covariates?...I believe they were factors in your model too?. I suggest to include the precalving value (only 1) as covariate. Similarly, probably, the parity nested in age and nested in the cow (parity/age/cow) must be considered the random factor. This will exclude most of your bias in your results. Have other smaller things, but would like to see this aspects modify first.
Answer: According to the reviewer recommendations, we have modified the statistical performance, that now can be read in lines 163-167: “Data for all measured variables (salivary analytes, WBC count, milk yield, and BCS) were analyzed as repeated measures using the mixed linear model of SPSS in which time was considered as fixed factor, the BCS and the nested interaction on parity × age as random factors, and the BCS at T-13 as covariate.”
When we made this new approach, most of the previous analyte results that showed significant changes kept their significance, and also interestingly, additional metabolic analytes showed significant changes such as urea, triglycerides, glucose, and lactate. This new information can be read in the “Results” section in lines 186-233.
Additionally, the “Discussion” section was modified concerning the new results obtained (lines 293-295, 302-311, and 312-321).
Reviewer 2 Report
Abstract-Simple Summary: Abstract is well written and is an excellent "mini-version of the manuscript.
Introduction: The well-referenced introduction contains pertinent and current information. It provides the necessary background to enable the reader to understand the subject the manuscript is addressing (Changes in saliva analytes in peripartum dairy cows).
Materials and methods: This section does a good job explaining the materials and methods utilized in this project. How did the authors arrive at the number of cows to sample (13)? Were these cows screened for Bovine Viral Diarrhea Virus, Bovine Leukosis Virus, or any other disease in the region that could impact immunity, milk production, or general metabolism? Line 78-80 indicates the animals were healthy, but how was health measured?
Results: Figure 1 and Tables 1-3 are easy to follow and conveys the results.
Discussion: The discussion is excellent, short, and on point. Recognized are the study's limitations (few animals, all from one dairy and one breed, and the need for more research).
Conclusion: Appropriate and concise.
Author Response
Reviewer 2
Abstract-Simple Summary: Abstract is well written and is an excellent "mini-version of the manuscript.
Introduction: The well-referenced introduction contains pertinent and current information. It provides the necessary background to enable the reader to understand the subject the manuscript is addressing (Changes in saliva analytes in peripartum dairy cows).
Results: Figure 1 and Tables 1-3 are easy to follow and conveys the results.
Discussion: The discussion is excellent, short, and on point. Recognized are the study's limitations (few animals, all from one dairy and one breed, and the need for more research).
Conclusion: Appropriate and concise.
Answer: These authors want to give the thanks for these kinds words concerning our study.
Materials and methods: This section does a good job explaining the materials and methods utilized in this project. How did the authors arrive at the number of cows to sample (13)?
Answer: The authors used the results from our previous sialochemistry study in dairy cows to decide the cow's number for this research [9], adding three additional cows just if any of the cows should be removed for situations such as disease. This new information can now be read in lines 85-89: “Thirteen multiparous Holstein-Friesian dairy cows (mean age = 4.7 ± 1.50; mean parity = 3.2 ± 1.46, min = 2, max = 7) in their last phase of gestation from a Spain commercial dairy herd (38º2’ N, 1º15’W) were retrospectively selected according to a BCS > 3 at -20 ± 6.91 days relative to calving. In a previous sialochemistry study, ten cows gave an adequate power [4].”
Were these cows screened for Bovine Viral Diarrhea Virus, Bovine Leukosis Virus, or any other disease in the region that could impact immunity, milk production, or general metabolism?
Answer: This farm is free of brucellosis, tuberculosis, Bovine Leukosis Virus, and pleuropneumonia. Additionally, the cows are vaccinated of Bovine Viral Diarrhea, Infectious Bovine Rhinotracheitis, Bovine parainfluenza-3, and Bovine respiratory syncytial virus. This new information was also added to the new version of the manuscript (lines 113-115): “The farm was free of brucellosis, tuberculosis, Bovine Leukosis Virus, and pleuropneumonia. The cows are vaccinated of Bovine Viral Diarrhea, Infectious Bovine Rhinotracheitis, Bovine parainfluenza-3, and Bovine respiratory syncytial virus.”
Line 78-80 indicates the animals were healthy, but how was health measured?
Answer: The authors have added the required reviewer’s information (lines 90-93): “An experienced veterinarian (PJV-M) checked all the cows, and no cows did show any health issues (i.e., mastitis, metritis, ketosis, or lameness) and had no hematological and serum biochemical abnormalities in routine profiles.”
Reviewer 3 Report
Well done work!
Material and methods are well described allowing the replication of the study, preliminary in its nature, as described in the title. It would be nice to have information on the handling of the saliva specimens before analysis through the automated analyzer (adding this important information would save time to readers by not having to search for it in other papers).
The results are well described too and discussed conclusively. Could the authors add literature on the relationship between the analyte levels in saliva and blood?
Authors write several times “…inside the 12 first hours after calving….”. The term “within” may be better understood than “inside”.
Page 8, lines 272-274: The sentence “Besides, it was observed increases in the retained placenta [33] and lameness [34] in the blood and mastitis in saliva” is difficult to understand. Do you mean that “increases of lactate concentration in the blood have been observed in cows with a retained placenta and lameness, and in saliva of cows with mastitis.”?
Author Response
Reviewer 3
Well done work!
Material and methods are well described allowing the replication of the study, preliminary in its nature, as described in the title. It would be nice to have information on the handling of the saliva specimens before analysis through the automated analyzer (adding this important information would save time to readers by not having to search for it in other papers).
Answer: We have included that information in the new version of the manuscript (lines 116-129): “The saliva sampling was performed by introducing a sponge clipped to a flexible thin metal rod into the cow’s mouth. The sponge was then placed in a collection tube Salivette (Sarstedt, Aktiengesellschaft & Co, Nümbrecht, Germany). After that, blood was obtained by venipuncture in the vena caudal, and it was stored in a heparin device (LH/Li Heparin, Aquisel, Barcelona, Spain). The saliva and blood were obtained in the milking parlor when the milkers were removed and the nipples were post-dipped [4]. Cows were previously made accustomed to the procedure for saliva and blood sampling. Both saliva and blood devices were stored less than two hours in ice until the processing: the saliva was centrifuged at 3.000×g for 10 min at 4 °C, and the blood analyzed for the white blood cells (WBC) count. Saliva specimens were stored less than five months at -80ºC until analysis.”
The results are well described too and discussed conclusively. Could the authors add literature on the relationship between the analyte levels in saliva and blood?
Answer: We have added literature concerning the current knowledge about the correlations between saliva and blood of some analytes (lines 345-347): “Also, it would be of interest to study the possible correlation between saliva and blood analytes, although, in a previous study in cows with mastitis, no evident correlations were found [5].”
Authors write several times “…inside the 12 first hours after calving….”. The term “within” may be better understood than “inside”.
Answer: The authors have replaced the term “inside” by “within” in all the document (lines 152, 194, 242, 265, and 391).
Page 8, lines 272-274: The sentence “Besides, it was observed increases in the retained placenta [33] and lameness [34] in the blood and mastitis in saliva” is difficult to understand. Do you mean that “increases of lactate concentration in the blood have been observed in cows with a retained placenta and lameness, and in saliva of cows with mastitis.”?
Answer: Thanks a lot for this suggestion. It has been included and can be read in the new version of the manuscript (lines 328-331).
Reviewer 4 Report
This ia an iteresting pilot study and a well written manuscript. However, major changes are needed as following:
Introduction: This is a poor introduction that does not correspond to the overall quality of the manuscript. The section must be re-rewritten to give more information (and references) about the welfare status of dairy cows at peripartum period and the methods applied to evaluate this.
You mention that 'saliva sampling could be an ideal method for stress and health evaluation...' but there is no evidence in your study about the changes in saliva analytes as related to health (inflammation, etc.). You are asked to put off any conclusions about the health of cows and strictly discuss the results of your study.
Material and Methods: You should explain the number of animals used in your study.
Author Response
Reviewer 4
This ia an iteresting pilot study and a well written manuscript. However, major changes are needed as following:
Introduction: This is a poor introduction that does not correspond to the overall quality of the manuscript. The section must be re-rewritten to give more information (and references) about the welfare status of dairy cows at peripartum period and the methods applied to evaluate this.
Answer: We have included in the “Introduction section” information about the welfare evaluation techniques in dairy cows (lines 60-74): “The peripartum period in dairy cows, which is comprised approximately 3 weeks before and after calving, has a direct impact on their subsequent health status, production, and profitability. This period is considered the highest risk for developing infectious diseases and metabolic disorders and, therefore, compromised welfare since they are submitted to a high metabolic challenge with significant physiological changes [6–8]. The welfare assessment in dairy cows can be made by behavioral (i.e., pain scales, stereotypes), physiological (i.e., heart rate variability), and performance (i.e., milk yield, fertility, dry matter intake, reduced body condition score -BCS-) parameters, but also by measuring in the blood hormone (i.e., cortisol, β-endorphin) and hematological and biochemical analytes (i.e., metabolites, leukocytes ratios, acute phase proteins, inflammatory products) [9]. However, to the best knowledge of these authors, no studies have been found where the possible changes in salivary analytes have been evaluated during the peripartum period in dairy cows to assess whether the possible salivary analytes changes are in line with the previous changes reported in blood.”
If the reviewer thinks that we should include more data or references, we would be happy to include it.
You mention that 'saliva sampling could be an ideal method for stress and health evaluation...' but there is no evidence in your study about the changes in saliva analytes as related to health (inflammation, etc.). You are asked to put off any conclusions about the health of cows and strictly discuss the results of your study.
Answer: We have adapted this sentence to this reviewer’s requirements and we have put off any conclusions about the health of cows. Therefore, we have deleted the sentence: “Therefore, saliva sampling could be considered an ideal method for stress and health evaluation in veterinary species”
Material and Methods: You should explain the number of animals used in your study.
Answer: We have modified the paragraph where we explain how the number of animals were selected (lines 85-89): “Thirteen multiparous Holstein-Friesian dairy cows (mean age = 4.7 ± 1.50; mean parity = 3.2 ± 1.46, min = 2, max = 7) in their last phase of gestation from a Spain commercial dairy herd (38º2’ N, 1º15’W) were retrospectively selected according to a BCS > 3 at -20 ± 6.91 days relative to calving. In a previous sialochemistry study, ten cows gave an adequate power [4].”
REFERENCES:
- Edmonson, A.J.; Lean, I.J.; Weaver, L.D.; Farver, T.; Webster, G. A Body Condition Scoring Chart for Holstein Dairy Cows. J. Dairy Sci. 1989, 72, 68–78.
- Bernabucci, U.; Ronchi, B.; Lacetera, N.; Nardone, A. Influence of Body Condition Score on Relationships Between Metabolic Status and Oxidative Stress in Periparturient Dairy Cows. J. Dairy Sci. 2010, 88, 2017–2026.
- Alharthi, A.; Zhou, Z.; Lopreiato, V.; Trevisi, E.; Loor, J.J. Body condition score prior to parturition is associated with plasma and adipose tissue biomarkers of lipid metabolism and inflammation in Holstein cows. J. Anim. Sci. Biotechnol. 2018, 9, 1–12.
- Paape, M.; Mehrzad, J.; Zhao, X.; Detilleux, J.; Burvenich, C. Defense of the bovine mammary gland by polymorphonuclear neutrophil leukocytes. J. Mammary Gland Biol. Neoplasia 2002, 7, 109–121.
- Burvenich, C.; Paape, M.J.; Hill, A.W.; Guidry, A.J.; Miller, R.H.; Heyneman, R.; Kremer, W.D.; Brand, A. Role of the neutrophil leucocyte in the local and systemic reactions during experimentally induced E. coli mastitis in cows immediately after calving. Vet. Q. 1994, 16, 45–50.
- Diez-Fraile, A.; Meyer, E.; Burvenich, C. Sympathoadrenal and immune system activation during the periparturient period and their association with bovine coliform mastitis. A review. Vet. Q. 2003, 25, 31–44.
- H. Zerbe, N. Schneider, W. Leibold, T. wensing, T.A.M. Kruip, H.J.S. Philic Granulocytes in Postpartum Cows Associated With Fatty Liver. Theriogenology 2000, 54, 771–786.
- Zachut, M.; Kra, G.; Nemes-Navon, N.; Ben-Aharon, N.; Moallem, U.; Lavon, Y.; Jacoby, S. Seasonal heat load is more potent than the degree of body weight loss in dysregulating immune function by reducing white blood cell populations and increasing inflammation in Holstein dairy cows. J. Dairy Sci. 2020, 103, 10809–10822.
- Contreras-Aguilar, M.D.; Vallejo-Mateo, P.J.; Želvytè, R.; Tecles, F.; Rubio, C.P. Changes in Saliva Analytes Associated with Lameness in Cows : A Pilot Study. Animals 2020, 10, 2078.
- Trevisi, E.; Bertoni, G. Some physiological and biochemical methods for acute and chronic stress evaluation in dairy cows. Ital. J. Anim. Sci. 2009, 8, 265–286.
Round 2
Reviewer 1 Report
Most of my concerns have been addressed.
Reviewer 4 Report
The authors addressed effectively the comments received.